# Platelet Innate Immune Receptors and TLRs: A Double-Edged Sword

**DOI:** 10.3390/ijms22157894

**Published:** 2021-07-23

**Authors:** Théo Ebermeyer, Fabrice Cognasse, Philippe Berthelot, Patrick Mismetti, Olivier Garraud, Hind Hamzeh-Cognasse

**Affiliations:** 1INSERM U1059-SAINBIOSE, Université de Lyon, F-42023 Saint-Etienne, France; theo.ebermeyer@univ-st-etienne.fr (T.E.); fabrice.cognasse@univ-st-etienne.fr (F.C.); Patrick.Mismettibis@chu-st-etienne.fr (P.M.); olivier.garraud@univ-st-etienne.fr (O.G.); 2Etablissement Français du Sang Auvergne-Rhône-Alpes, 25 bd Pasteur, F-42100 Saint-Étienne, France; 3Team GIMAP, CIRI—Centre International de Recherche en Infectiologie, Université de Lyon, U1111, UMR5308, F-69007 Lyon, France; philippe.berthelot@univ-st-etienne.fr; 4Infectious Diseases Department, CHU de St-Etienne, F-42055 Saint-Etienne, France; 5Department of Vascular Medicine and Therapeutics, INNOVTE, CHU de St-Etienne, F-42055 Saint-Etienne, France

**Keywords:** toll-like receptor, immunomodulatory molecules, immuno-surveillance, inflammation, pathophysiology

## Abstract

Platelets are hematopoietic cells whose main function has for a long time been considered to be the maintenance of vascular integrity. They have an essential role in the hemostatic response, but they also have functional capabilities that go far beyond it. This review will provide an overview of platelet functions. Indeed, stress signals may induce platelet apoptosis through proapoptotis or hemostasis receptors, necrosis, and even autophagy. Platelets also interact with immune cells and modulate immune responses in terms of activation, maturation, recruitment and cytokine secretion. This review will also show that platelets, thanks to their wide range of innate immune receptors, and in particular toll-like receptors, and can be considered sentinels actively participating in the immuno-surveillance of the body. We will discuss the diversity of platelet responses following the engagement of these receptors as well as the signaling pathways involved. Finally, we will show that while platelets contribute significantly, via their TLRs, to immune response and inflammation, these receptors also participate in the pathophysiological processes associated with various pathogens and diseases, including cancer and atherosclerosis.

## 1. Introduction

Platelets are the smallest non-nucleated circulating blood elements and are derived from megakaryocytes in the bone marrow [1]. Platelets primarily maintain vascular integrity and participate in the coagulation cascade at the vascular injury site [2]. Platelets are well-known to be central in primary hemostasis, adhering to the damaged vascular bed caused by subendothelial collagen exposure. Activated platelets aggregate and release specific mediators to strengthen thrombus formation, alongside other blood cells (i.e., erythrocytes and leukocytes), and prevent further bleeding [3].

A hundred billion platelets must be produced daily to maintain the platelet physiological level, estimated at 150–400 G/L, in circulation because of their short lifespan (between 8 and 10 days). Moreover, aged or damaged platelets are eliminated in the liver, spleen, and lung. Platelets act as blood sentinels of vascular integrity and thus prevent leakage and bleeding. Another important—though less commonly evoked—function is that platelets, in their role as sentinels, actively participate in anti-infectious responses. Thus, platelets indeed contribute to a regulated and physiologic thrombosis process, termed immunothrombosis, upon systemic infection [4]. Platelets and immune cells collaborate during this process to form a physical barrier preventing pathogen escape and leading to the activation of the innate and adaptive immune responses. In addition, platelets orchestrate the immune response by modulating several immune cells (e.g., neutrophils [5], lymphocytes, monocytes, and dendritic cells [6]).

To exert these functions, platelets constitutively express a plethora of immune-sensing receptors, including not only nod-like receptors [7], C-type lectins [8] families, and toll-like receptors (TLRs), but also the FcγRIIA receptor (also known as CD32a), which binds immune complexes [9]. These receptors sense highly conserved exogenous pathogen-associated or endogenous damage-associated molecular patterns (PAMPs and DAMPs, respectively). In turn, pathogen linkage triggers platelet activation and the release of soluble immunomodulatory factors. Platelets are indeed heavily loaded in the immunomodulatory factors contained in alpha granules, dense granules, and other organelles, which may all—though differentially, depending on the stimuli—be mobilized very quickly upon the reception of a danger signal. Moreover, their secretion is operated by the open canalicular system (OCS), which enables their efficient and fast release in the extracellular milieu. In addition, platelets may also release a huge variety of immunomodulatory molecules through the shedding of membrane proteins [10].

However, platelets do not only contribute to the resolution of bleeding and infection, but also at the nexus of the development and evolution of cardiovascular and associated diseases (e.g., infective endocarditis [11], atherosclerosis [12], spreading of cancer metastasis [13], and sepsis [14]). The pathophysiology of the thromboembolic events that occur as a result of severe acute respiratory syndrome coronavirus 2 (SARS-CoV-2)-mediated disease (coronavirus disease 2019; COVID-19) seems to involve platelets. A reduced platelet count [15], a prolonged prothrombin time [16], and an alteration in platelet responses [17,18] have been shown resulting from COVID-19. However, the basic mechanisms underpinning infection-associated coagulopathies are still poorly understood.

This review aims to revisit the functions of platelet innate immune receptors, represented by TLRs (and others), in immune physiological and pathophysiological processes, illustrating the two functions of the platelets that can contribute to both pathogenic mechanisms and their resolution.

## 2. The Central Role of Platelets in Hemostasis

Platelets are known to be particularly active in maintaining hemostasis so as to prevent, control, and stop hemorrhages. Platelet hemostasis is a multistep process that involves the concerted action of several membrane receptors and intracellular signaling pathways.

In the absence of a vascular breach, platelets are kept in a quiescent state in the bloodstream because of the inhibitory pathways involving the prostacyclin (PGI2) receptor and receptors with immuno-tyrosine-based inhibition motifs (ITIM), such as PECAM (Figure 1). In steady-state conditions, the vascular endothelium converts arachidonic acid into PGI2 by use of type 1 and 2 cyclooxygenases and prostacyclin synthase. Moreover, PGI2 induces the production of intracellular cyclic AMP (cAMP), inhibiting platelet function [19]. Similarly, nitric oxide inhibits platelet functions by inducing the production and accumulation of cyclic GMP (cGMP) within platelets [20]. Elevated levels of cAMP and cGMP regulate the activity of the proteins kinase A (PKA) and PKG, which, in turn, phosphorylate the molecules involved in platelet inhibition. However, upon injury to the vessel wall, platelets spread and adhere to the extracellular matrix molecules, including collagen fibers, fibronectin, and von Willebrand factor (vWF). Platelets bind to collagen either directly, through their GPVI or GPIaIIa (i.e., α2/β1 integrin) receptors, or indirectly, via the vWF bound to collagen, which is then sensed by the GPIb-V-IX complex on platelets [21]. Furthermore, the C-type lectin-like receptor 2 (CLEC-2), which interacts with podoplanin and is expressed by many cell types but absent from vascular endothelial cells and platelets membranes [22], mediates platelet activation through Syk and phospholipase C (PLC) γ2 signaling with 1,2-diacylglycerol (DAG) and inositol 1,4,5-trisphosphosphate (IP3) generation that leads to calcium mobilization [23]. Platelets may also bind to the fibrinogen adsorbed onto the injured vascular wall through the GPIIbIIIa receptors (i.e., αIIbβ3 integrin) [24].

In addition, several soluble agonists, including adenosine diphosphate (ADP), thromboxane A2 (TxA2), thrombin, and serotonin, are secreted by activated platelets themselves or by other cell types, and may also fuel platelet activation. ADP binds to its P2 purinergic receptors, namely P2Y1 and P2Y12, and the activation of the former leads to the mobilization of intracellular calcium via PKC activation, and that of the latter to the inhibition of cAMP production and the activation of Phospho-Inositide 3-kinase, altogether inducing platelet activation. TxA2, resulting from the metabolism of arachidonic acid by cyclooxygenase, binds to its TP receptor and also promotes the mobilization of intracellular calcium, as does serotonin, upon binding to its 5-HT2A receptor [25]. Galectin-1 and -8 also induce platelet activation via intracellular calcium mobilization after direct binding to platelets or via the GPIb-V-IX complex as extracellular matrix proteins [26]. Finally, thrombin is the most potent platelet activator and acts through the protease-activated receptor (PAR) family, which is composed of three members. While PAR-1 and PAR-3 are exclusively expressed by human and murine platelets, respectively, PAR-4 expression is common to both [27]. Thrombin is cleaved by PAR-1 or PAR-4, and the active peptides (Ser-Phe-Leu-Leu-Arg-Asn, SFLLRN and Gly-Tyr-Pro-Gly-Lys-Phe, and GYPGKF for PAR-1 and PAR-4, respectively) form a complex that activates the PAR-associated G protein cascade. Although PAR-1 has a higher affinity than PAR-4 for thrombin, the peptide obtained upon cleavage by PAR-4 produces a stronger signal than that obtained after cleavage by PAR-1. Moreover, the simultaneous inhibition of both PAR-1 and PAR-4 prevents any aggregation in vitro, even at high thrombin concentrations [28]. A PAR-4 deficiency in mice renders platelets insensitive to thrombin stimulation and protects them from thrombosis [29].

GPIIbIIIa changes conformation, from an inactive low- to a high-affinity form reactive to fibrinogen and vWF, after initial platelet activation. This operation promotes platelet spreading and aggregation. Those interactions lead to changes in platelet shape and trigger the release of granule contents. This results in the autocrine enhancement of primary aggregation, and then in the formation of thrombi. Platelet aggregation is followed by secondary hemostasis processes, which aim to stabilize the platelet plug. The resulting coagulation cascade, through either the extrinsic pathway involving tissue factor or the intrinsic pathway via contact activation, leads to the generation of thrombin, which plays a central role. Indeed, thrombin reinforces the coagulation cascade, catalyzes fibrin generation from insoluble fibrinogen, and also induces Factor XIII activation to form Factor XIIIa, which crosslinks fibrin fibers, stabilizing the thrombus.

Aside from hemostasis, platelets secrete and use the secretome in inflammation pathways. Thrombin can also activate the PAR-1 on endothelial cells to trigger the production of CCL2 and interleukin (IL)-6 [30,31], and provoke the mobilization of the CD40 ligand (CD40-L) and CD62P (i.e., P-selectin) to the platelet surface [32]. Moreover, CD62P mediates platelet adhesion to monocytes, neutrophils, and endothelial cells. Finally, platelet activation induces a rearrangement of the phospholipids in the plasma membrane via the exposure of phosphatidylserines (PS) to the surface of the platelets, which is a marker of platelet apoptosis and has a procoagulant effect [33].

With a dual role in hemostasis and inflammation, platelets bridge hemostasis, inflammation, and anti-infectious responses. Thus, their activation and apoptosis states are of particular importance in physiological processes as well as under stressful circumstances.

## 3. Platelet Response to Stress

### 3.1. Apoptosis

Apoptosis is a process of programmed cell death, whereby cells initiate self-destruction in response to an extra- or intracellular trigger. The physiological mechanism of apoptosis controls the lifespan of cells, including aspects of tissue structure conservation or tissue repair. However, this mechanism is generally considered to apply to nucleated cells [34]. That said, features of apoptosis pathways have also been described in platelets, e.g., the activation of the intrinsic mitochondrial pathway, or the extrinsic pathway involving *Death Domains* receptors followed by highly regulated caspase activation cascades [35]. Thus, apoptosis delimits the lifespan of platelets in the circulation [36], and is associated with the storage lesion of platelet concentrates, which limits their shelf life [37].

Platelet apoptosis has been shown to result in cytoplasmic condensation, PS exposure, and CD62P expression on the platelet plasma membrane, the integrity of which is, however, retained, and functional platelets are eliminated by phagocytosis [38]. Moreover, PS externalization is a common feature during apoptosis [33] and triggers cell elimination [36]. Both human and murine platelets generally display increased PS exposition upon aging [39]. Furthermore, mitochondria dysfunction is also found, and is characterized by altered mitochondrial signaling pathways, inner mitochondrial membrane disruption, and cytochrome c leakage.

In addition, PKA activity seems to be correlated with platelet survival. Indeed, a significant reduction in PKA activity has been observed in stored platelets, as well as in vitro-maintained aged platelets (obtained upon 16 h incubation at 37 °C of platelet-rich plasma) [40] and in platelets isolated from patients with immune thrombocytopenia, diabetes, and sepsis. Moreover, PKA inhibition results in the sequestration of prosurvival protein Bcl-xL in mitochondria and subsequent apoptosis, while PKA activation protects platelets from apoptosis [41].

Moreover, Siglec-7, a surface lectin belonging to the immunoglobulin superfamily and a sialic acid receptor, is involved in the regulation of human platelet apoptosis through four key pathways: (1) the depolarization of the mitochondrial inner membrane, (2) the overexpression of proapoptotic Bcl-2-associated X (Bax) and Bcl-2 homologous antagonist killer (Bak) proteins and decreased expression of anti-apoptotic Bcl-2 proteins, (3) abnormal PS exposure to the platelet membrane, and (4) the formation of platelet microparticles [42], which could promote both the coagulation and the apoptosis of surrounding cells [43,44].

In addition, human platelets were found to express some *classical* apoptosis receptors [45] and adaptor proteins, including tumor necrosis factor (TNF)-associated factor (TRAF)-2/5, Fas-associated protein with death domain (FADD), and death effector domain-associated factor (DEDAF) [46]. Interestingly, treating platelets with ligands of the receptors involved in extrinsic apoptosis pathways, such as TNF, Fas Ligand, and TWEAK, did not induce caspase 3/7 activation or PS exposure in this study, suggesting different triggering machinery for the extrinsic pathway. Moreover, extrinsic apoptosis may occur independently from death receptors through platelet receptors usually involved in the hemostatic response [47]. Indeed, platelet activation and apoptosis seem to be related, because platelet apoptosis can be mediated via PAR-1, GPIIb/IIIa, and GPIbα receptors, among many other stimuli [48]. However, activation and apoptosis can simultaneously occur, but are driven by independent pathways according to the stimulus [49].

Finally, accumulating evidence indicates that platelet activation and apoptosis result in thrombocytopenia or an increased cardiovascular risk in several infectious diseases [50,51,52]. For instance, platelet stimulation induced by a peptidoglycan monomer prepared from a *Staphylococcus aureus* 113 strain activates the caspase-3 protein, which is common to both the extrinsic and intrinsic pathways of apoptosis and involves TLR2 [53]. Similarly, Kraemer et al. also demonstrated that uropathogenic bacteria, isolated from a septic patient, induces apoptotic events in human platelets by degrading Bcl-xL, an important protein regulating cell survival [54].

### 3.2. Necrosis

Necrosis is characterized by an alteration of the cytoskeleton and the plasma membrane, leading to the loss of permeability of the latter. Necrosis may follow apoptosis and therefore be secondary. However, it can also occur independently from apoptosis and become primary. Primary and secondary eukaryotic cells display many different features. While secondary necrotic cells are small, lose their chromatin, and release nucleosome-bound high-mobility group protein B1 (HMGB1), primary apoptotic cells are larger, retain their chromatin, and release free HMGB1 in the extracellular milieu [55]. However, both types of necrotic cells externalize PS.

Regarding anucleate platelets, distinguishing necrosis from apoptosis appears to be difficult because classical methods using DNA intercalating agents to assess plasma membrane permeability cannot be used. This challenge has been addressed by different teams, either by developing new specific platelet necrosis markers [56] or by relying on cyclophilin D (CypD) involvement in the necrotic process [57] (Table 1). Indeed, necrosis pathways involve the formation of CypD-dependent mitochondrial permeability transition pores (MPTP) in the inner mitochondria membrane [58]. Moreover, MPTP formation is essentially regulated by CypD because a CypD deficiency alters platelet response in mice [59].

Necrotic platelets have been shown to actively contribute to thrombosis in a CypD-dependent manner by displaying a procoagulant surface [56,60], associating to neutrophils and promoting their accumulation in the brain in a mouse cerebral ischemia–reperfusion injury model [61]. A similar mechanism has been described upon gut ischemia in mice, with an accumulation of aggregates comprised of neutrophils and necrotic platelets in the lungs promoting thrombosis, which could be improved by using mice deficient in CypD, specifically in the platelets [57].

In addition to thrombosis, necrosis may also contribute to inducing an inflammatory response. Unlike apoptosis, which is associated with *eat-me* signals to promote their clearance and induce anti-inflammatory responses, necrosis and particularly secondary necrosis promote the recruitment of proinflammatory phagocytes by releasing DAMPs [62], including HMGB1–nucleosome complexes [55]. HMGB1, which exerts a potent inflammatory role through TLR2 and TLR4 [63], is released by platelets, and illustrates the central role of platelets in bridging stress as well thrombotic and immune responses [64,65].

Finally, necroptosis, another regulated cell death mechanism, may also occur in platelets. Necroptotic cells have a necrotic-related morphotype and involve a signaling pathway requiring receptor-interacting protein kinase 3 (RIPK3) and mixed lineage kinase domain-like proteins. Both have been reported as effective in platelets and involve thrombus formation and stability [66,67]. Interestingly, as observed for HMGB1, RIPK3 also seems to stand at the interface of inflammatory and stress responses [68], and would deserve further investigation in platelets.

### 3.3. Autophagy

Autophagy is a regulated biological process responsible for cytoprotection via the specific elimination of unnecessary or dysfunctional organelles and cellular components. Moreover, it also provides nutriments and energy upon short-term cell starvation. However, it can also lead to cell death because autophagic and apoptotic pathways are tightly intricated and regulate each other [69]. Several types of autophagy have been described, mainly microautophagy, chaperone-mediated autophagy, and macroautophagy. Micro- and macroautophagy differ according to the molecular mechanisms involved, but share a common step consisting of protein degradation by the lysosome [70]. In addition, microautophagy involves lysosomal invagination entrapping a small fraction of the cytoplasm, followed by protein degradation. Chaperone-mediated autophagy requires the binding of chaperone-proteins, such as heat shock proteins, onto the corresponding substrate to be catabolized, which is then transferred to the lysosome-associated membrane protein type 2A (LAMP-2A), translocated within the lysosome, and then hydrolyzed. Macroautophagy consists of successive steps involving several proteins, including autophagy-related proteins (ATG), leading to autophagosome formation. Briefly, ATG101 and ATG14L associate with other proteins of the nucleation complex to initiate the elongation step of the phagophore membrane. In parallel, ATG3, 4, and 7 contribute to the production of the lipophilic LC3-II protein from LC3-I, which is derived from its precursor pro-LC3. Consequently, LC3-II is then bound to both sides of the phagophore membrane by an ATG5/ATG12 complex as the elongation of the phagophore proceeds, and therefore constitutes a relevant autophagy marker. The phagosome closes in on itself and encloses part of the cytoplasm, and then fuses with the lysosome, leading to autophagosome formation, the degradation of its content, and the recycling of the catabolized products. This pathway can be activated in response to different types of stress situations (e.g., nutrient starvation) [71].

Functional autophagy machinery were detected for the first time a few years ago in platelets, because autophagy was efficiently induced by starvation or mechanistic target of rapamycin (mTOR) inhibition, which is a negative regulator of autophagy [72] (Table 1). These authors also demonstrated that autophagy inhibition prevents platelets from adhering and aggregating upon collagen or thrombin stimulation. The interconnections between platelet activation and autophagy pathways have been further examined, and Ouseph et al. elegantly showed that the autophagic flux, reflected by LC3-II levels, was altered upon platelet activation, proportional to the strength of the stimulus [73]. It was also shown to involve platelet activation signaling molecules (e.g., intracellular calcium, PKC, or PLC). In addition, they also showed in ATG7-defective murine platelets the crucial role of this molecule in platelet aggregation, but not in platelet count, fibrinogen binding, or platelet spreading, leading to impaired hemostasis and thrombus formation in vivo. Similarly, another molecule involved in autophagy processes, vacuolar protein sorting 34 (VPS34, also known as class III PI3K), was shown to contribute to platelet aggregation, but was not intrinsic to platelets’ adhesion onto collagen or their spreading, because thrombus formation was altered in VPS34-deficient mice, while GPIIbIIIa activation and downstream signaling remained unaffected [74]. Therefore, autophagy regulation may be of importance in pathologies affecting platelet functions. For instance, platelets from diabetic patients display apoptotic markers and have damaged mitochondria, which undergo significant mitophagy, as depicted by ultrastructural changes and the overexpression of several autophagy proteins [75]. However, this phenomenon could be beneficial for diabetic patients because increased oxidative stress and subsequent apoptosis are observed in both human and mouse platelets deficient in the mitophagy-related protein Parkin upon mitophagy inhibition. Similarly, a study on platelets from immune thrombocytopenia patients showed that these platelets had lower levels of the LC3-II protein, which could be rescued by ex vivo supplementation with the 6-amino-2,3-dihydro-3-hydroxymethyl-1,4-benzoxazine autophagy enhancer, leading to decreased platelet apoptosis and improved platelet viability [76]. Finally, recent work by Tan et al. evidenced an autophagy increase, regulated by the PI3K/AKT/mTOR pathway, upon storage of apheresis platelets, which was accompanied by increased phenotypic platelet activation and a decreased platelet aggregation rate [77]. Apheresis platelet autophagy was also negatively correlated with the efficacy of clinical platelet transfusion assessed 24 h post-transfusion, suggesting that increased autophagy in apheresis related to platelet storage may affect transfusion efficiency [77]. Altogether, these observations make autophagy an interesting marker of platelet physiology, particularly upon aging (apoptotic, necrotic and autophagic platelets responses are summarized in Table 2).

## 4. Platelets as Immune Cells

Platelets can be considered immune cells in several regards. The first fact that supports this statement is that they are filled with secretory granules containing, along with hemostatic factors, a collection of immunomodulatory and antibacterial molecules [78,79].

### 4.1. Release of Platelet Immunomodulatory and Antimicrobial Soluble Factors

After activation, platelets trigger the exocytosis of their granule content due to surface-connected invaginations of the OCS, which intervenes in both the delivery of platelet contents to the extracellular milieu and the uptake of circulating elements [80].

The first type of platelet secretory granules is alpha granules, which contain a vast array of proteins, including membrane-bound adhesive receptors (e.g., GPVI, GPIIb/IIIa, or the GPIb-IX-V complex). These granules also enclose vWF, fibrinogen, fibronectin, and vitronectin, as well as growth factors (e.g., insulin-like growth factor (IGF), transforming growth factor-beta (TGF-β), and platelet-derived growth factor (PDGF)) and numerous immunomodulatory molecules and chemokines (e.g., CCL3/MIP-1α, CCL5/RANTES, and CXCL4/PF4) [81].

Interestingly, Italiano et al. determined, using specific immune electron microscopy, the presence of a specific subset of platelet alpha granules containing angiogenic molecules at the ultrastructural level [82]. Moreover, these distinct subpopulations of alpha granules may undergo selective release under diverse platelet stimuli, i.e., the engagement of different signaling pathways, as was reported by Chatterjee et al. [83].

Platelets also contain dense granules, filled with ADP, serotonin, and calcium. All three are involved in blood coagulation and express CD63 and LAMP-1/2, which are also present in lysosomal granules. Platelet lysosomes store acid hydrolases that are involved in extracellular matrix component degradation and receptor cleavage, but could also contribute to the cytosolic elements in autophagic processes [84]. Moreover, Thon et al. identified new T granules that contain TLR9, which specifically recognizes bacterial DNA CpG sequences, highlighting the active role of platelets in innate responses to pathogens [85]. Finally, the circulating forms of CD40L and CD62P molecules expressed by activated platelets, resulting from membrane cleavage, also contribute to leukocyte activation through their receptors, CD40 and P-selectin glycoprotein ligand-1 (PSGL-1), respectively, which could lead to inflammatory complications [86,87]. Thus, the secretory response of platelets has several implications for the immune response. On one hand, it can contribute directly to the elimination of pathogens due to antibacterial molecules. On the other hand, it can also activate and modulate the leukocyte response [88].

Furthermore, platelets store in their granules several chemokines that have antimicrobial properties, including CXCL4 and CCL5, as well as thrombocidins and kinocidins, which are C-terminal peptides of CXC chemokines, human beta-defensins-1, 2, and 3, and thymosin-β4 [89]. All of these can potentially be released upon platelet activation. Indeed, in vitro thrombin-stimulated human platelets release several peptides (e.g., RANTES, CXCL4, or thymosin-β4) with antimicrobial activities against *Escherichia coli* or *S. aureus* [90]. In addition, platelet microbicidal proteins (PMPs) and thrombin-stimulated PMPs have also been shown to be highly effective against bacteria [91]. Additionally, CXCL4, which is abundantly produced by platelets, appears to be a critical modulator of viral replication. Indeed, this chemokine was shown to enhance dengue virus 2 in vitro replication in monocytes and impede interferon production by infected cells [92]. In contrast, platelet CXL4 inhibited the replication of several primary and reference strains of the human immunodeficiency virus (HIV)-1 by blocking not only the early stages of the viral replication, but also the attachment of HIV-1 gp120 to CD4, its main entry receptor, on target cells [93]. Platelet CXCL4 was also shown to kill *Plasmodium falciparum* merozoites in erythrocytes by lysing the parasite’s digestive vacuole [94]. However, the opposite effect of CXCL4 has been observed in HIV-1 infection of macrophages, where it potentiated viral replication [95], demonstrating that one factor could exert contrasting effects according to the model considered.

### 4.2. Modulation of Immune Cell Response

In addition to their well-established role in hemostasis, platelets participate in the host immune response by communicating with leukocytes either directly through contact interactions or indirectly via the secretion of immunomodulatory and polarizing molecules. Activated platelets can form complexes with leukocytes (monocytes, neutrophils, natural killer cells, and even eosinophils) during the progression of various inflammatory situations [96,97]. Zuchtriegel et al., using intravital microscopy, reported that critical inflammation promotes the rapid adherence of platelets to endothelial cell junctions through the interaction of platelet GPIIb/IIIa with the vWF expressed within these junctions. This step is concomitant to the formation of heterotypic aggregates with neutrophils or inflammatory monocytes. These cell–cell interactions rely on both CD62P/PSGL-1 and CD40/CD40L couples and promote the recruitment of leukocytes from the bloodstream to inflammatory sites [98].

#### 4.2.1. Platelet/Neutrophil Interactions

Neutrophils are at the frontline of immune defense due to their enzyme-rich secretory and phagocytic capacity. Moreover, neutrophils—known to be central to control—infection can also contribute to the pathogenic processes. Indeed, dysregulated neutrophilic responses may be harmful to host tissues in several autoimmune diseases [99].

During inflammation, the CD62P expressed on activated platelets and endothelial cells initiates tethering to leukocytes via the PSGL-1 receptor. This interaction is negatively regulated by the release of the neutrophil proteinases cathepsin G and elastase, which cleave PSGL-1 from the neutrophil surface, an operation that abrogates cell–cell interaction with platelets and endothelial cells through CD62P [100]. Thus, PSGL-1 engagement on the neutrophil surface leads to downstream signaling via mitogen-activated protein kinase (MAPK), and also stimulates IL-8 release [101]. Moreover, platelet transmembrane glycoprotein junctional adhesion molecule 3 (JAM-3) has been shown to directly bind to the leukocyte integrin Mac1 (CD11b/CD18) [102]. Another such tandem is formed by platelet GPIbα [103].

Platelets play an important role in promoting neutrophil recruitment to inflammatory sites by forming platelet–neutrophil aggregates that circulate in the blood and that can migrate out of the vasculature, reach the periphery, and release enzymes and inflammatory factors in infiltrated tissues. The formation inhibition of these aggregates abrogated neutrophil migration in experimental mice inflammation models [104,105]. This cooperation between platelets and neutrophils relies, at least in part, on platelet P-Rex/Vav guanine–nucleotide exchange factors [106].

The formation of neutrophil extracellular traps (NETs), by which neutrophils extrude a chromatin-based structure that acts as an extracellular scaffold to capture and kill microorganisms, requires close cooperation between platelets and neutrophils [107]. Hence, platelet activation induced by lipopolysaccharide (LPS) sensing by TLR4 has been reported to induce platelet binding to neutrophils. This results in the formation of robust NETs under flow conditions, accompanied by the release of matrix metalloproteinase 9 (MMP-9), an enzyme that is capable of destroying connective tissue and causing persistent inflammation. Furthermore, plasma from septic patients, but not from healthy volunteers, added in vitro to platelets recovered from healthy persons induced platelet–neutrophil binding as efficiently as high levels of LPS did.

Another relevant platelet-derived intermediary of NET formation is human β-defensin 1. This peptide is secreted by platelets following exposure to *S. aureus* α-toxin and may induce NET formation without the requirement of the direct contact of platelets with neutrophils [79].

However, cellular activation during platelet–neutrophil interactions is reciprocal. Activated neutrophils release antimicrobial peptides (e.g., α-defensins and cathelicidins), which also have chemotactic properties for eosinophils and neutrophils. Cathelicidin LL-37, for instance, promotes platelet activation via the GPVI integrin with downstream tyrosine kinases Src/Syk and phospholipase C signaling, which, in turn, fuels platelet–neutrophil interactions [108].

#### 4.2.2. Platelet/Monocyte Interactions

Platelets have been shown to modulate the effector functions of monocytes. Indeed, activated platelets polarize the inflammatory monocyte response by inducing the activation of nuclear factor-κB (NF-κB) and the synthesis of proinflammatory mediators IL-8 and CCL-2 [109]. However, activated platelets also promoted an anti-inflammatory response in monocytes exposed to *E. coli* LPS or *Porphyromonas gingivalis* stimulation, consisting of IL-10 production and a reduction in TNF-α production [110]. Moreover, platelets directly interact with monocytes, inducing enhanced expressions of CD40, PSGL-1, and CD11b on their surface [111]. This further enhances platelet–monocyte aggregate formation and the recruitment of monocytes to the endothelium via CCL-2 secretion [112].

Moreover, platelet-associated HMGB1 promotes monocyte recruitment through receptors for advanced glycation end-products, and prevents monocytes from experiencing apoptosis via the TLR4-dependent activation of the MAPK/ERK pathway in monocytes [65].

#### 4.2.3. Platelets/Dendritic Cell Interactions

Dendritic cells (DCs) are professional antigen-presenting cells in humans, contributing to the initiation and polarization of innate and adaptative immune responses. DCs further interact with platelets. Moreover, Czapiga et al. demonstrated that platelet-derived CD40L helps to mature DCs into the functional presentation of antigens against T lymphocytes via the upregulation of costimulatory molecules and IL-12/p40 production [113]. Our group, as well as others, has shown that platelets can regulate DC maturation, either through direct contact or via their secretory products [6,114]. In addition, activated platelets have been shown to improve bacteria uptake and killing by dendritic cells due to sCD40L [115]. However, a recent study showed that platelet-soluble factors reduced the proinflammatory response of preactivated DCs while promoting an anti-inflammatory response, preventing the polarization of autologous naïve T cells into a Th1 phenotype [116]. These contradictory observations highlight the importance of considering the microenvironment when examining platelet and leukocyte relations, including the prior activation state of each cell type upon their mutual contact.

#### 4.2.4. Platelets/B and T Cell Interactions

Platelets interact with the adaptive immune system by attracting and stimulating T and B cells [117]. Thus, in an experimental model of cerebral malaria in mice, platelets are activated by *Plasmodium*-infected red blood cells and release CXCL4, which accounts for T cell recruitment and activation. Moreover, T cell trafficking to the brain was significantly reduced in CXCL4-deficient mice [118]. Interestingly, class I major histocompatibility complex (MHC-I) molecules have been observed on platelets. Moreover, thrombin-activated platelets could present antigens in MHC-I and efficiently stimulate antigen-specific T cells, including the murine model of cerebral malaria [119]. Very recently, in a murine model of polymicrobial sepsis, platelets were shown to upregulate MHC-I expression, and to internalize, process, and present antigens to CD8^+^ T cells, which resulted in a decrease in their proliferation and IFNγ production [120]. However, the specific deletion of MHC-I in the platelet lineage restored CD8^+^ T cell responses and improved mice survival, suggesting that platelets contribute to the impaired immune response during sepsis by suppressing cytotoxic responses, at least in mice models.

Moreover, platelets activate B cells via the release of soluble CD40 and CD40L, leading to proliferation, differentiation, isotype switching, and memory B cell generation [121]. Moreover, a study by Sprague et al. showed that wild-type platelet transfusion in CD40L-deficient mice during viral infection significantly increased circulating levels of virus-specific immunoglobulin G (IgG), providing further evidence of the link between platelets and the adaptive immune system [122].

However, the contribution of platelets to the immune response is not limited to the secretion of immunomodulatory molecules or immune cell regulation (summarized in Table 2). Indeed, platelets also express innate immune receptors (e.g., TLRs, C-type lectin receptors, integrins, and G protein-coupled receptors), allowing them to detect PAMPs and DAMPs.

### 4.3. Platelets as Pathogen Sensors

Platelets are abundant and highly sensitive to variations in their environment, and have been compared to drones that watch the circulation from an immunological as well as a hemostatic point of view [123]. To efficiently perform their immunosurveillance function, they have a large array of receptors that allow them to detect any invading pathogen or endogenous danger signal, ranging from the IgG receptor FcγRIIa to pattern recognition receptors (PRRs), and a large family of various innate immunity receptors, including TLRs (the latter will be explored in the next section).

FcγRIIa and plasma Ig appear to be involved in platelet activation in response to the H1N1 virus, which results in 12-HETE and microparticle release [124]. In addition, platelets bind immunocomplexes containing bacteria. Consequently, complexes induce platelet aggregation through FcγRIIa and IgG signaling [125,126]. This operation promotes bacteria killing [127], although other studies failed to show the implication of FcγRIIa in *S. aureus* elimination by platelets [128]. This mechanism has been shown to promote *E. coli* killing by platelets, involve the FcγRIIa receptor, be amplified by platelet PF4, and require αIIbβ3 integrin activation [129].

In addition to its contribution to the Fc-mediated bacteria killing by platelets, the αIIbβ3 integrin has been shown to mediate platelet interaction with bacteria, either directly through several bacterial proteins or indirectly by involving plasma proteins (e.g., fibrinogen) [130]. Similarly, GPIb may also directly bind some bacterial components (e.g., GspB and Hsa streptococcal adhesins [131] or the serine-rich protein A), leading to platelet activation.

Several platelet PRRs are involved in pathogen recognition; for instance, C-type lectins comprised of the dendritic cell-specific intercellular adhesion molecule-3–grabbing nonintegrin (DC-SIGN) and C-type lectin-like receptor 2 (CLEC-2) receptors (Figure 2). Both these receptors bind carbohydrate moieties of PAMPs through their conserved carbohydrate recognition domains. Moreover, Simon et al. demonstrated that DC-SIGN, with the help of heparan sulfate proteoglycans, enables platelets to specifically recognize and internalize the dengue virus, a process that was enhanced upon thrombin stimulation [132]. Platelets become activated and undergo apoptotic processes that rely on DC-SIGN as a blockade of this receptor upon exposure to dengue virus, preventing platelet apoptosis [133]. In addition, DC-SIGN expression is decreased on platelets from severe dengue virus-infected patients [134], probably as a result of platelet apoptosis inducing a partial depletion of the platelet subpopulation expressing DC-SIGN. Furthermore, platelet CLEC-2 has also been shown to contribute to dengue pathophysiology. Thus, the binding of the dengue virus on platelet CLEC-2 induced platelet activation and promoted the release of extracellular vesicles that significantly contribute to inflammation and NET formation [135]. Finally, the contribution of CLEC-2 to HIV-1 binding on platelets, along with DC-SIGN, has also been evidenced, and both receptors contribute to the efficient transmission of viral particles from platelets to T cells [8].

Another type of PRR that appears to contribute to pathogen sensing by platelets is NOD leucine-rich repeat and pyrin domain-containing protein 3 (NLRP3). Indeed, upon contact with several types of infectious pathogens, platelet NLRP3 becomes activated via to mechanisms not fully elucidated yet. For instance, in mice sepsis models, the activation of NLRP3 in platelets and an increase in caspase-1 activity were noted. This was translated into systemic and peripheral inflammation, endothelial leakage, and organ damage, all reversed at least in part when using a selective NLRP3 inhibitor [136]. Similarly, NLRP3 activation in platelets was attributed to excessive reactive oxygen species (ROS) production by mitochondria upon dengue virus infection. This sequence led to caspase-1 activation and IL-1β secretion, a consequence of which was the alteration of vascular permeability [7]. A recent work by Quirino-Teixeira et al. further suggested that the dengue virus nonstructural protein-1 (NS1) could be translated within the platelets upon infection. This induces NLRP3-driven IL1β production with secretion in the extracellular milieu, where it could activate platelet TLR4 [137].

All of the above illustrates the intertwining of the mechanisms involved in pathogen recognition by PRRs, and the complexity of studying them in platelets.

**Figure 2 ijms-22-07894-f002:**
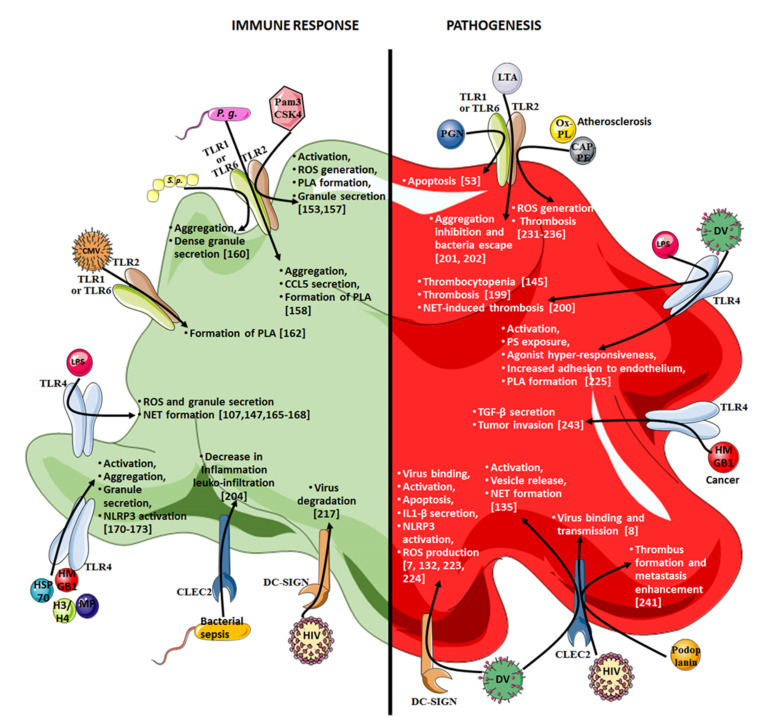
Platelet membrane innate immunity and toll-like receptors involved in the immune response and pathogenesis. CAP-PE: carboxyalkylpyrrole-phosphatidylethanolamine, CLEC2: C-type lectin-like receptor 2, DC-SIGN: dendritic cell-specific intercellular adhesion molecule-3–grabbing nonintegrin, DV: dengue virus, H3/H4: H3 and H4 histone, HIV: human immunodeficiency virus, HMGB1: high-mobility group box 1, HSP: heat shock protein, IL: interleukine, LPS: lipopolysaccharide, LTA: lipoteichoic acid, MP: microparticle, NET: neutrophil extracellular trap, Ox-PL: oxidized phospholipid, P.g.: porphyromonas gingivalis, PGN: peptidoglycan, PLA: platelet-leukocyte aggregate, PS: phospatidylserine, ROS: reactive oxygen species, TLR: toll-like receptor.

### 4.4. A Focus on Platelet TLRs

TLRs were first described in 1988 by Hashimoto et al. in *Drosophila melanogaster* [138], followed by the discovery of human homologs (e.g., TLR4) several years later [139]. TLRs detect PAMPs from pathogens as diverse as bacteria, viruses, fungi, and parasites, and have been described on circulating as well as noncirculating cells (e.g., epithelial cells) [140]. The mammalian TLR family comprises 10 members in humans (TLR1-10), with two additional TLRs in mice (TLR1-9 and TLR11-13) [141]. Mammalian TLRs are localized either on the cell surface or in intracellular compartments (e.g., endoplasmic reticulum, endosome, lysosome, and so on), and discriminate between self and nonself-moieties [142]. About 15 years ago, our group, along with others, described the presence and functionality of TLRs in mice and human platelets, on their membranes (TLR2/TLR1/TLR6/TLR4) and intracellularly (TLR3, TLR7, and TLR9) [143,144,145].

### 4.5. TLR Structure

TLRs are type I transmembrane proteins composed of extracellular leucine-rich repeat (LRR) motifs that mediate the recognition of PAMPs, transmembrane domains, and cytosolic Toll IL-1 receptor (TIR) endodomains that interact with the downstream adaptors required for signaling [146]. The LRR ectodomains consist of 19–25 tandem sequences containing leucine, each of which is under 30 amino acids in length [140]. Furthermore, each LRR motif consists of a β strand and α helix linked by loops.

According to the composition of their subunits, TLRs operate as heterodimers or homodimers. TLRs undergo conformational changes following ligand binding. For instance, TLR2 forms heterodimers with either TLR1 or TLR6 for pathogen recognition, while others are homodimerized.

### 4.6. TLR Signaling Pathways in Platelets

The engagement of TLRs by infectious PAMPs triggers the activation of signaling cascades, leading to the induction of genes involved in antimicrobial host defense. After ligand binding, TLRs undergo dimerization and conformational changes that induce the recruitment of TIR domain-containing adaptor proteins (TIRAP) to the TIR domain of the TLR. 

The main step in the TLR signaling pathway shared by all TLRs is the MyD88-dependent activation of NF-κB, as observed by our group, along with the groups of Shattner and Blumberg and Phipps [147,148,149]. In addition, TLR signaling is mediated by MyD88 for all TLRs. Remarkably, TLR4 can signal through either the MyD88-dependent or MyD88-independent signaling pathway in platelets [147]. The MyD88-dependent pathway involves IRAK-1, IRAK-4, and TRAF-6 molecules leading to PI3K/Akt or ERK1/2 activation, while the MyD88-independent pathway is engaged upon TLR4 internalization and requires toll receptor-associated activator of interferon (TRIF), TRAF-3, and IKK factors, although this has not been definitely confirmed [150].

NF-κB is one of the many regulators of inflammation-related genes that are induced upon TLR triggering. Interestingly, although non-nucleated, platelets express common NF-κB family members, including IκB and NF-κB [149,151]. NF-κB signaling in platelets involves IKKβ phosphorylation, IκBα degradation, and p65 phosphorylation [152]. Moreover, NF-κB inhibition in platelets is involved in their hemostatic response, as NF-κB reduces platelet responses to hemostatic stimuli (e.g., collagen [152] or thrombin [153]). These observations highlight the central role of NF-κB in platelets in anti-infectious and hemostatic responses, opening the path to new remedial perspectives of thromboinflammatory diseases.

Moreover, other molecules are implicated in the signaling pathways associated with platelet intracellular TLRs. Indeed, TLR3 signaling relies on the TRIF adaptor protein. TRIF activation promotes the formation of a complex between TRAF-3- and IKK-related kinases, which activate interferon regulatory factor (IRF)-3. The triggering of MyD88 or TRIF leads to the activation of several nuclear factors, including NF-κB, AP-1, and IRF-3, which result in the release of inflammatory cytokines and type I interferons [154]. Furthermore, TLR7 activation in platelets involves Akt and p38-MAPK phosphorylation, leading to the membrane expression of CD62P [155]. However, unusual TLR9 ligands, referred to as carboxyethylpyrrole protein adducts (CAPs), promote platelet activation via the MyD88 pathway [156].

### 4.7. Expression and Functional Response of Platelet TLRs

#### 4.7.1. TLR2

TLR2 has been identified on human and mouse platelets, and forms a heterodimer with TLR1 or TLR6 to recognize bacterial lipopeptides such as Pam3CSK4, a synthetic triacylated lipopeptide used as a TLR1/2 ligand (Figure 2). Indeed, Pam3CSK4 induces human platelet activation and ROS generation, and increases binding to neutrophils in whole-blood assays via the PI3k-Akt pathway [157]. *S. aureus* peptidoglycan has also been shown to induce platelet activation leading to apoptosis [53].

Moreover, TLR2 has been reported to mediate platelet aggregation and CCL5 secretion in response to *P. gingivalis* [158]. *P. gingivalis* infection increased circulating platelet–neutrophil aggregates in the whole blood of wild-type but not TLR2-deficient mice [157]. In addition, TLR2 triggering by Pam3CSK4 has been shown to promote platelets to produce a significant inflammatory response with the release of sCD62P, CCL5, and sCD40L, which was inhibited by a selective blocking-TLR2 antibody [153]. However, although earlier studies failed to evidence any platelet activation upon Pam3CSK4 stimulation, which could result from variations in experimental protocols and reagents used [159], it is now acknowledged that platelet TLR2 is functional and involved in the response to pathogens. Furthermore, Keane et al. have evidenced that *Streptococcus pneumoniae*-induced platelet aggregation is strain-specific, but relies on neither capsular serotype nor any molecules secreted from *S. pneumonia*. Moreover, the authors have demonstrated that *S. pneumonia* binding to platelet TLR2 activates the PI3K/RAP1 pathway, which results in dense granule release [160]. However, in another study, *S. pneumonia* failed to induce platelet aggregation, although it did cause platelet phenotypical activation independently from TLR2 [161]. The same study also showed that, upon *S. pneumonia* nasal infection, the platelets of wild-type mice did not release a larger number of inflammatory factors than those from mice knocked out specifically in the megakaryocytic lineage. This apparent discrepancy in the involvement of platelet TLR2 in response to *S. pneumonia* may be explained by the conditions of the interaction of platelets with the pathogen. Thus, the in vitro stimulation of platelets by *S. pneumonia* strains differed. While Kean et al. used 7 × 10^9^ colony-forming units (CFUs) of *S. pneumonia* to stimulate platelet, de Stoppelaar used 5 × 10^7^ viable CFUs, which could have an effect on the receptor engagement because the ratio between platelets and bacteria is a hundred times lower and would preferentially occur through other, more abundant receptors (e.g., integrins). Moreover, as mice were infected with streptococci by the nasal route, it is highly likely that the bacterium first makes contact with pneumocytes and alveolar macrophages. This may affect platelets for the second time by modifying their microenvironment and favoring other platelet receptors and signaling pathways.

Finally, human cytomegalovirus (CMV) has been shown to preferentially bind to TLR2 subsets of platelets. This observation was confirmed by studying murine platelets lacking TLR2, which did not respond to murine CMV stimulation [162]. Moreover, human CMV-activated platelets rapidly formed complexes with leukocytes, involving CD62P receptors on platelets and PSGL-1 on leukocytes, through TLR2 signaling [162].

#### 4.7.2. TLR4

TLR4 is the most extensively studied molecule of the TLR family. It is the sensing receptor for LPS, the lipid A of which is responsible for endotoxin activity. The membrane-bound lipopolysaccharide-binding protein (LBP) and the opsonic receptor CD14 were found to be active partners for the ligation of LPS to host cells [163]. This multistep recognition process is initiated by the binding of LPS to the LBP that carries LPS to CD14, and the complex is then delivered to the myeloid differentiation 2 (MD-2)–TLR4 complex. The CD14 receptor can be found directly on the plasma membranes of some cells, or, as is the case with platelets, which do not express CD14, at the plasma level in a soluble form [164]. Upon stimulation by LPS, platelets have been shown to activate and produce significant amounts of a variety of immunomodulatory factors. For instance, upon TLR4-triggering by LPS, platelets have been shown to produce IL-1β-bearing inflammatory microparticles through kinase activation leading to NF-κB activation, which illustrates, on one hand, the platelet contribution to inflammatory responses, and, on the other hand, the nongenomic role of NF-κB in platelets (Figure 2) [165]. The TLR4-stimulation of platelets with LPS from *E. coli* and *Pseudomonas aeruginosa* also induced ROS production that could be antagonized by a β-d-glucan via two mechanisms: direct antioxidation, and competition with LPS for binding to the TLR4–MD-2 complex [166].

Moreover, platelets appear to be able to discriminate between various isoforms of LPS that differ slightly in their structures. The team of the current study showed that the in vitro stimulation of platelets with LPS from either *E. coli* O111 or *Salmonella minnesota*, which are smooth and rough isoforms of LPS, respectively, induces the differential release by platelets of immunomodulatory molecules that then trigger a differential secretion of IL-6, TNF-α, and IL-8 by peripheral blood mononuclear cells [147]. This was confirmed by Kappelmayer et al., who observed that the rough form of *S. minnesota* LPS (Re-LPS) was more likely to enhance platelet activation than smooth *E. coli* LPS [167]. In addition, various LPSs from enterohemorrhagic *E. coli* serotypes (O103, O111, O121, and O157) presented different abilities to bind to platelets, compared to the nonenterohemorrhagic O111:B4 serotypes at a similar dose, which confirmed the capacity of platelet TLR4 to finely discriminate between closely related molecules [168].

However, apart from LPS, platelet TLR4 recognizes a collection of ligands displaying various structures, including fibronectin [169], heat shock proteins HSP70 [170], histones H3 and H4 [171], microparticles [172], and HMGB1 [173], which all induce efficient platelet responses (e.g., CD62P overexpression, secretion of diverse immunomodulatory factors, induction of clotting, and NLRP3 activation).

Platelet TLR4 engagement by bacteria was shown to affect platelet interactions with leukocytes, in particular neutrophils. Moreover, Clark et al. have proposed that the activation of neutrophils by platelets occurs upon TLR4 engagement by LPS on platelets, leading to the formation of NETs that can ensnare bacteria under shear forces [107]. Moreover, the same study showed that plasma from individuals presenting with bacterial sepsis engendered TLR4-mediated NET formation. However, the infection does not seem to regulate TLR expression on platelets because platelets from infected patients showed an activation pattern (increased CD62P, CD63, and phosphatidylserine exposure), while TLR expression was not influenced. Similarly, LPS stimulation did not induce TLR overexpression by platelets, while hemostatic stimulation with a thrombin analog (TRAP) did [174].

#### 4.7.3. TLR3

The recognition of double-stranded RNA (dsRNA) by TLR3 has been shown in TLR3-deficient mice, which showed a reduced inflammatory response to poly(I:C), a synthetic dsRNA analog (Figure 3) [175]. Platelets have been shown to express TLR3 intracellularly (16%) rather than on their surface (8%) under resting conditions, while activation with thrombin significantly increased surface TLR3 expression (41%) [176]. Another study evidenced the presence of TLR3 in megakaryocytes and platelets, and has shown the involvement of PI3K/Akt, ERK1/2, and NF-κB pathways in megakaryocytes and platelets upon TLR3-triggering [177]. Moreover, TLR3-engagement by poly(I:C) has been shown to enhance the expression of CD62P by platelets, thus increasing the intracellular calcium concentration and causing the secretion of CXCL4 after 30 min of stimulation. IL-1β secretion required longer stimulation [177]. However, another set of experiments suggested that the poly(I:C) stimulation of platelets did not induce platelet aggregation or activation [178]. Thus, TLR3-stimulation could potentiate platelet aggregation upon hemostatic stimulation, rather than inducing aggregation by itself, as evidenced by D’Atri et al. [177]. Moreover, TLR3 may be implicated in respiratory syncytial virus (RSV) internalization because platelets exposed to RSV showed an activated phenotype, although the involvement of other receptors could not be ruled out [179]. Finally, in an interesting way, the expression of TLR transcripts in platelets has been suggested to be associated with gender, TLR3 being significantly more expressed in females. TLR3 expression also correlated with several cardiovascular risk factors and the elevated presence of inflammatory factors in the circulation. However, platelet TLR4 and TLR2 transcripts were more frequently associated with inflammatory biomarkers than TLR3 [180].

#### 4.7.4. TLR7

Koupenova et al. described functional TLR7 on both human and murine platelets [155]. Interestingly, in contrast to TLR7-deficient mice, wild-type mice infected with the encephalomyocarditis virus (EMCV) or exposed to the TLR7 agonist loxoribine underwent thrombocytopenia (more pronounced upon loxoribine exposure). TLR7 activation also resulted in an increased CD62P expression by platelets via p38-MAPK and Akt pathways. The enhanced CD62P expression by platelets promoted their interaction with neutrophils, leading to platelet internalization by neutrophils. This mechanism could therefore contribute to thrombocytopenia associated with EMCV. However, despite its role in thrombocytopenia, platelet TLR7 activation improved mice survival, as platelet–granulocyte aggregate formation enabled ECMV clearance. In addition, TLR7 stimulation may also induce a hypersensitive phenotype in platelets, because platelets exposed to nucleic acids were shown to be more responsive to low doses of thrombin in a TLR7/8/9-dependent way [181].

Finally, another study by the Freedman group has recently confirmed the role of TLR7 as a regulator of platelet–neutrophil interactions. This study evidenced the internalization of influenza particles by platelets, resulting in C3 release in a TLR7-dependent way, as a selective antagonist of TLR7 abrogated C3 excretion [182]. Platelet-derived C3 was able to induce neutrophil activation and the release of DNA. In turn, activated neutrophils would promote the release of GM-CSF by platelets, which would reduce the effect of C3, thus establishing a negative feedback loop in neutrophil–platelet activation.

Furthermore, the role of platelet TLR7 also extends to the regulation of platelet activation during interaction with viruses. Thus, the post-binding platelet response to HIV, after VAMP-8- and Arf6-dependent endocytosis, was recently demonstrated to involve TLR7. A colocalization of HIV particles and TLR7 has been evidenced in compartments also expressing LC3-associated phagosomes [183]. Moreover, platelet activation upon HIV endocytosis required compartment acidification and dynamin, a GTPase involved in clathrin-mediated endocytosis, as did TLR7 triggering by loxoribin. Finally, platelet activation upon HIV endocytosis occurred via a MyD88-, Akt-, IRAK4- and IKK-dependent pathway, and resulted in an inflammatory platelet response, including granule secretion and the formation of platelet–leukocyte complexes.

#### 4.7.5. TLR9

Platelet TLR9 expression was observed by our team in 2005, and was detected mostly in the cytoplasm (34%), with a little membrane expression (10%). Upon platelet activation with thrombin, a significant modulation of TLR9 expression was observed, both on the surface and intracellularly, in CD41+/CD62P+ cells (Figure 3) [144], which was not the case upon LPS stimulation [184]. Moreover, Thon et al. evidenced that TLR9 is preferentially retained in specific intracellular vesicles, named T granules. After exposure to multiple agonists (e.g., thrombin, phorbol myristate acetate, collagen-related peptide, ADP, or type IV collagen), T granules colocalize with vesicle-associated membrane protein 8 upon platelet spreading, and TLR9 molecules are presented to the cell surface [85].

Furthermore, classical ligands of TLR9 are synthetic oligodeoxynucleotides (ODNs) that contain unmethylated CpG sequences, more frequently found in bacterial genomes compared to mammalians [185]. CAPs are oxidative endogenous products that represent additional intracellular TLR9 ligands that activate platelets via the TLR9/MyD88 signaling pathway, inferred from the fact that CAPs failed to induce platelet activation in MyD88-deficient mice [156]. Interestingly, platelets from patients with acute coronary syndrome expressed higher levels of TLR9, which suggests a higher sensitivity of platelets from these patients to TLR9 agonists [186]. Furthermore, platelets from antineutrophil cytoplasmic antibody-associated vasculitis have been shown to express higher levels of CD62P and TLR9, which are significantly associated with lung lesions [187]. The same study demonstrated that CpG ODNs stimulation prompted platelets to interact with neutrophils, which resulted in increased NET formation via the release of CXCL4 by platelets in a CpG ODN dose-dependent manner.

#### 4.7.6. TLR5, 8, and 10

Few studies have investigated platelet TLR5, 8, and 10. Moreover, mRNA transcripts were first evidenced in a large cohort of participants in the Framingham Heart Study [180]. Thereafter, TLR5 expression on platelets was found to be increased upon thrombin activation and in patients with bacterial sepsis [174]. Furthermore, TLR8 mRNA has recently been found to be increased in platelets upon stimulation with *Candida*-derived chitin, while other TLR mRNA was not modulated, which suggests a role in platelet responses to *Candida* [188].

## 5. Platelet Contribution to Pathogenesis through Their Innate Immunity Receptors

Due to their wide range of innate immune receptors, and their ability to secrete immunomodulatory molecules and interact with many immune cells, platelets contribute to the resolution of infection. However, depending on the context, platelet determinants can also contribute to several pathophysiological mechanisms (e.g., inflammation, propagation of infection, and metastasis spreading. For an overview, refer to Figure 2 and Figure 3).

### 5.1. Sepsis

In 2016, the third international consensus SEPSIS-3 redefined sepsis as a life-threatening organ dysfunction caused by a dysregulated host response to infection [189]. Sepsis is a general inflammatory response to harmful infections, which eventually leads to tissue necrosis, multiple organ failure, and mortality. This pathology is associated with high mortality because the mortality rate is around 25–30% in sepsis patients [190], and could exceed 40% in cases of septic shock [189]. The deregulation of both inflammation and hemostasis processes during sepsis suggests an active contribution of platelets to sepsis pathophysiology, through the innate immunity receptors. Thus, the activation of platelet TLR4 by LPS was reported to be responsible for the copious inflammatory factor release [191] and reduced platelet counts (at least in a mouse model of endotoxemia) [145]. Thrombocytopenia is associated with increased mortality in critically ill patients, as observed by Claushuis et al. in a large cohort of 931 critically ill patients with sepsis [192]. HMGB1, another TLR4 ligand, is an important inflammatory mediator in lethal sepsis [193]. Moreover, increased circulating levels of HMGB1 have been described in experimental sepsis models [194], as well as in patients with bacterial sepsis [195]. HMGB1 could modulate platelet responses in addition to other circulating cells. In addition, sepsis may also be associated with an alteration of platelet responsiveness to stimulation because it has been observed that platelets from patients with sepsis, in contrast to healthy controls, did not upregulate TLR4 expression upon thrombin stimulation, which suggests an impairment of platelets’ capacity to sense danger upon the establishment of sepsis [196]. However, the expression levels of TLR4 on platelets during sepsis are discussed in studies showing unaltered TLR expression on platelets from sepsis patients [174], while others describe increased TLR4 expression on sepsis platelets [197]. This is of particular importance because platelet TLR4 is probably involved in the thrombotic response of platelets to sepsis, even independently of MyD88 signaling [198], and platelet TLR4 signaling may use other pathways, including TRIF [150,151]. The occurrence of thrombotic complications was reduced in an endotoxemia model induced in TLR4-deficient mice. This reduction was reversed, and thrombotic complications were more frequent, when the deficient mice were transfused with platelets originating from wild-type animals [199].

The role of platelet TLRs during sepsis could be even more ambivalent, as illustrated by the involvement of platelets in the formation of NETs, which, on one hand, cause thrombosis [200], and, on the other hand, trap and help to eliminate bacteria [107]. Similarly, lipoteichoic acid, a membrane component of Gram-positive bacteria, was shown to inhibit platelet aggregation in human platelets [201] and reduce platelet thrombus formation in vitro [202], which may not cause thrombotic complications, but could be of advantage to the bacteria that would not be trapped in a thrombus. In addition, TLR3 and TLR4 could be of particular importance in platelet responses during sepsis, because a deficiency in either of these receptors is associated with more severe thrombocytopenia during polymicrobial sepsis induced by a cecal ligation and puncture procedure, while TLR2- and TLR7-deficient mice displayed a less severe coagulopathy, suggesting a different pattern of TLR involvement in murine polymicrobial sepsis [203]. Finally, another innate immunity receptor, C-type lectin-like-2/CLEC-2, found on platelets could be beneficial during sepsis by downregulating inflammation and leukocyte infiltration in tissues and reducing organ damage, further illustrating that the contribution of platelets to sepsis pathophysiology is not unequivocal [204].

### 5.2. COVID-19

COVID-19, caused by SARS-CoV-2, is associated with a high in-hospital mortality rate related to the severity of pneumonia and the development of systemic complications, including thromboinflammatory events [205], which suggests the active participation of platelets. Moreover, low platelet counts are associated with increased risk of severe outcomes in COVID-19-infected individuals [206], and a platelet-hyperactivated, apoptotic, and procoagulant/prothrombotic phenotype has been reported in COVID-19 infection, partly induced by antibody/FcγRIIa interactions [207,208,209]. In addition to displaying this pathogenic phenotype, platelets from severe COVID-19 patients display huge modifications in their transcriptome, including transcripts involved in mitochondrial function and stress proteins (e.g., HMGB1) [210], and form aggregates with circulating monocytes, inducing tissue factor expression in the latter [211]. COVID-19 platelets have also been shown to release abundant proinflammatory molecules, which contribute to the cytokine storm [18], as well as extracellular vesicles that worsen thrombosis [212]. Moreover, how platelets interact with SARS-CoV-2 remains to be clarified because studies report conflicting results regarding the presence of the SARS-CoV-2 receptors angiotensin-converting enzyme 2 (ACE2) and transmembrane protease serine 2 (TMPRSS2), as some authors evidenced the involvement of both receptors in platelets [213], while others failed to detect them [18,210,214], suggesting that alternative receptors may also be involved in virus entry into platelets, as has been shown at the RNA and ultrastructural levels [18,213].

### 5.3. HIV-1 Infection

Platelets also act as immune cells when it comes to binding and internalizing viruses, including influenza and HIV, due to their TLRs [182,183]. HIV is part of the lentivirus family, composed of two species, HIV-1 and HIV-2, and it infects cells of the immune system (helper T cells, macrophages, and dendritic cells). The first observation of platelets’ ability to internalize HIV-1 particles was initially made by Zucker-Franklin et al. [215], and was confirmed and clarified later on. Moreover, Youssefian et al. have shown that HIV-1 particle engulfment by platelets occurs in vitro in specific subcellular compartments, and that uptake of this virus can also occur in vivo [216]. CLEC-2 and DC-SIGN have been proposed by Chaipan et al. to be involved in HIV-1 particles’ interactions with platelets, as the use of inhibitors for both CLEC-2 and DC-SIGN resulted in a strong reduction in HIV-1 association with platelets [8]. More recently, the contribution of TLR7 to HIV-1 pseudovirus uptake has also been evidenced, and was shown to involve MyD88 and IKK signaling and to lead to platelet activation and CXCL4 secretion [183]. However, the ability of platelets to transfer HIV particles could not be assessed in this study, which used nonreplicative pseudoviruses. Indeed, the issue of whether viral particles are internalized or not by platelets is of primary importance, because it appears to be related to the ability of platelets to transmit an infectious virus. Lentiviral vectors expressing an HIV-1 envelope have been shown to retain ultrastructural integrity in endocytic vesicles near the plasma membrane upon internalization into platelets via the DC-SIGN receptor, but to be degraded upon vesicle trafficking to the surface-connected canalicular system [217]. In contrast, in the study by Chaipan et al., platelets pulsed with HIV-1 were able to transfer infectious HIV-1 particles to target cells up to 48 h after initial viral interaction, a process that involved, at least in part, DC-SIGN and CLEC-2, although HIV-1 internalization was not investigated [8]. Similarly, platelets have recently been shown to efficiently transfer HIV-1 virions by forming heterotypic complexes with CD4+ T cells, which were all the more effective as they were strongly activated [218]. Interestingly, activated platelets overexpressed CLEC-2, but not DC-SIGN, which could account for the increased viral transmission. However, HIV-1 internalization has not been assessed either in this study.

As observed for bacteria, platelets can discriminate between viral patterns and produce an adapted secretory response. Thus, the in vitro stimulation of healthy platelets with several peptides from the gp120 and gp41 glycoproteins of the HIV-MN strain resulted in platelet activation and significant CCL5 production, which differed according to the peptide used for stimulation, while CXCL4 secretion was not affected [219]. However, the molecular basis of such a subtle distinction between close structures remains to be clarified. Platelet activation is indeed a recurrent parameter observed in platelets from people living with HIV, or upon the in vitro infection of healthy platelets. In addition, the alteration of platelets’ mitochondrial inflammatory response [220], sometimes with platelet exhaustion [52,221], and the increased release of inflammatory microparticles [222] make platelets central contributors to the HIV-1-related chronic inflammation that increases the risk of cardiovascular complications.

### 5.4. Dengue

The DC-SIGN receptor involved in platelet interactions with HIV-1 is also implicated in dengue virus detection. In addition, the use of anti-DC-SIGN monoclonal antibodies drastically reduced the binding of dengue virus to platelets, as did heparin, demonstrating the involvement of DC-SIGN and heparan sulfate proteoglycan in viral attachment to platelets. This was enhanced upon platelet activation as a consequence of probable receptor upregulation (though this was not assessed in the study) [132]. Moreover, due to the abundance of the intracellular organelles involved in post-transcriptional and translational processes, including ribosomes, rough endoplasmic reticulum, Golgi and mitochondria, platelets were shown in the same study to support efficient dengue virus replication, resulting in the production of lytic neo-virions, prevented by the use of a translational inhibitor. However, this did not occur without an alteration in platelet function. Thus, platelets from dengue-infected patients display increased activation and apoptotic features that correlate with disease severity [133]. Furthermore, a concomitant production of IL-1β was noted [7]. The ability of the dengue virus to cause mitochondrial dysfunction and apoptosis via DC-SIGN was confirmed in vitro on healthy donors’ platelets. Moreover, NLRP3 inflammasome activation has also been evidenced in platelets from dengue-infected patients and from healthy controls exposed in vitro to the virus, which results from the increased ROS production by platelet mitochondria (via the RIP-1/RIP-3 kinase pathway) and leads to vascular leakage and monocyte activation [7,223]. Very recently, NLRP3 activation was confirmed in platelets upon infection with dengue virus, and the involvement of the envelope protein domain III has been evidenced as an important inducer of platelet pyroptosis related to NLRP3 activation [224]. In addition to DC-SIGN, CLEC-2 has been demonstrated to participate in platelet activation by dengue virus, as reflected not only by the expression of activation markers, but also by the release of extracellular vesicles capable of activating neutrophils, causing the formation of NETs, and increasing the rate of mouse death. The specific involvement of CLEC-2 was verified by the use of an antibody blocking CLEC-2 or platelets from mice deficient in this receptor, as neither condition allowed platelet activation by the dengue virus or the production of extracellular vesicles [135]. Finally, the interaction of the dengue virus has also been evidenced with another innate immunity receptor on platelets, TLR4. Thus, the nonstructural protein 1 (NS1) of the dengue virus has been shown to trigger CD62P expression and PS exposure on platelets, both of which were reduced upon the treatment of platelets with a TLR4 signaling inhibitor or an anti-TLR4 blocking antibody [225]. The dengue NS1 protein, after activation, induced a hyperreactive state in the platelets, leading them to aggregate in response to suboptimal doses of ADP, to adhere more easily to HUVEC endothelial cells, and to form complexes with THP1 murine macrophages before being internalized by them. Taken together, these observations illustrate, for a single pathogen, the diversity of platelet responses and mechanisms involved, depending on the platelet receptor and the viral antigen sensed.

### 5.5. Atherosclerosis

Activated platelets display a variety of receptors and release multiple inflammatory mediators, which have been described to be involved in the formation of atherosclerotic lesions. For instance, platelets CXCL4 and CCL5 have been shown to form stable heterodimers whose disruption prevents atherosclerosis in ApoE-deficient atherosclerotic mice [226]. The role of platelet sCD62P atherogenesis has also been evidenced in the ApoE-deficient mouse model, because mice receiving CD62P-deficient platelets developed less atherosclerosis than mice receiving CD62P expression [227]. In addition, Lutgens and Daemen have shown that CD40L inhibition, also in the ApoE-deficient mouse model, resulted in the development of stable atherosclerotic plaques [228]. Platelet CD40 has also been shown to promote atherosclerosis by stimulating leukocyte and endothelial cell activation. Moreover, Gerdes et al. recently demonstrated that platelet-induced atherosclerosis progression was attenuated by the repeated in vivo injection of platelets from CD40- and ApoE-deficient mice [229]. Moreover, ApoE-deficient mice lacking either MyD88 or TLR4 exhibited reduced aortic atherosclerosis, lower levels of proinflammatory IL-12 and CCL2, as well as lower macrophage infiltration in aortic sinus plaques, suggesting the role of innate immunity and danger signal receptors in the pathophysiology of atherosclerosis [230].

Thus, plasma levels of carboxyalkylpyrrole–phosphatidylethanolamine (CAP-PE), derived from polyunsaturated fatty acid oxidation, are increased in hyperlipidemic ApoE-deficient mice, and induce in vitro platelet activation in a TLR2/1-dependent manner, as the use of blocking antibodies and the competitive TLR2 ligand prevented platelet activation. Moreover, CAP-PE promoted TLR2-dependent thrombosis in vivo because TLR2 deficiency improved the CAP-PE-induced occlusion of the carotid artery in mice [231]. Furthermore, the same group showed that, in platelets, TLR2 cooperated with TLR6 and the scavenger receptor CD36 to sense oxidized phospholipids in ApoE-deficient mice and enhance thrombosis through the MyD88/TRAF6 pathway [232].

The contribution of platelet CD36 to hyperlipidemia-related thrombosis has previously been suggested, because the genetic deletion of CD36 in ApoE-deficient mice reduced the platelet hyperreactivity that promoted thrombosis [233]. The investigation of human platelet activation upon in vitro exposure to oxidized phospholipids and atherosclerosis development in ApoE-deficient mice suggested that the platelet CD36 signaling pathway is involved in ROS accumulation through NAD phosphate oxidase 2, leading to platelet activation, insensitivity to PGI2, and thrombus formation [234,235,236]. Finally, investigating the contribution of platelet TLR7 to the pathogenesis of atherosclerosis may be interesting because TLR7 has been suggested to participate in the development of atherosclerosis in ApoE-deficient mice [237], and because platelet TLR7 stimulation leads to the development of platelet–neutrophils aggregates [155].

### 5.6. Cancer

The pathogenesis of cancer involves a complex microenvironment comprised of pro- and anti-inflammatory factors, angiogenic molecules, and several cell types, including platelets, which may be key players. Thus, cancer is associated with a hypercoagulable state that promotes venous thromboembolism and arterial thrombosis [238,239], suggesting the active role of platelets in its pathogenesis. For instance, platelet-derived microparticles have been suggested to play a role in cancer cell proliferation, angiogenesis, and inflammation [240]. In addition, platelet CLEC-2, whose ligand podoplanin is expressed on the surface of tumor cells, may facilitate tumor metastasis. Thus, via a CLEC-2-depleted mouse model, Shirai et al. showed the involvement of CLEC-2 in not only metastasis and thrombus formation in the lungs, but also in tumor growth, by enhancing tumor cell proliferation [241]. Thus, the formation of platelet aggregates would shelter tumor cells and promote their growth and metastatic dissemination. Similarly, platelet GPVI may also contribute to metastasis processes, as it has been shown to bind galectin-3 on tumor cells. Upon GPVI–galectin-3 interaction, platelets express CD62P, release ATP, and migrate across the endothelium, favoring the transmigration of tumor cells, which could be reduced via GPVI blockade [242]. Moreover, Yu et al. demonstrated a mechanism by which platelets support tumor cell metastasis, involving TLR4 and its endogenous ligand HMGB1 [243]. The same study also showed that platelets from TLR4-deficient mice secreted less TGF-β and were less efficient in promoting tumor invasion. Similar results were obtained when HMGB1 was neutralized, demonstrating the importance of the TLR4–HMGB1 axis in the protumor role of platelets.

## 6. Conclusions and Future Perspectives

Platelets have functions that extend beyond hemostasis. Good-quality evidence exists that shows that platelets are at the crossroads of thrombosis and inflammation, and that their innate immunity receptors, including TLRs, link these two central processes by responding to endogenous as well as to exogenous danger signals. TLR triggering initiates a complex signaling pathway cascade, which is further interconnected with those of the thrombotic response, resulting in a multifaceted platelet response. Platelets induce phenotypic and secretory activation, interact with the endothelium and immune cells, and contribute to the inflammatory response and subsequent repair process. However, depending on not only their microenvironment, including the cytokine environment and the activation status of their cellular partners, but also on the nature of the trigger, platelets may contribute to the deregulated inflammatory responses and immune escape of the causative agents, whether microbial or tumoral. Therefore, further investigations of platelets’ contribution to pathologies with inflammation would be of particular importance. Thus, the central place of platelets, their abundance, and their active participation in several key responses to biostress, make them candidates to regulate these processes, either to boost them when they are not efficient enough, or to slow them down when they contribute to chronic inflammation or a pathogenic phenomenon.

## Figures and Tables

**Figure 1 ijms-22-07894-f001:**
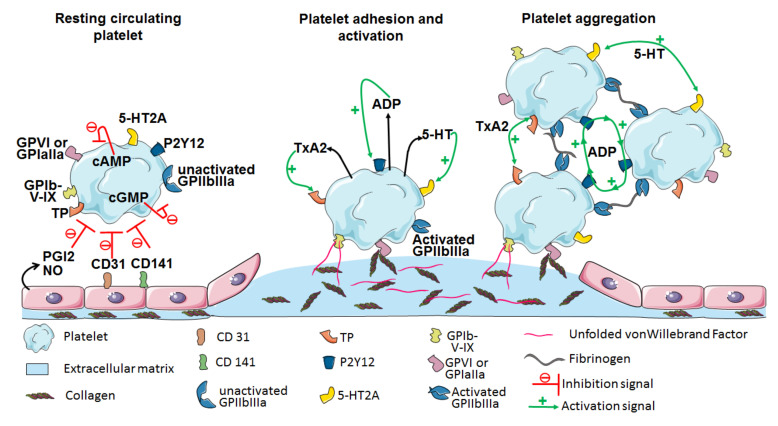
Platelets’ central role in hemostasis. Under resting conditions, platelets circulate in the vessel and are kept in a quiescent state by unaltered endothelial cells that express CD31 (i.e., Platelet endothelial cell adhesion molecule-1, PECAM-1) and CD141 (i.e., thrombomodulin) and secrete prostacyclin (PGI2) and nitric oxide (NO). PGI2 and NO induce the platelet production and intracellular accumulation of cyclic AMP (cAMP) and cyclic GMP (cGMP), respectively, which are both involved in platelet inhibition. Upon vascular disruption and the exposure of the underlying extracellular matrix, platelets bind to unfolded von Willebrand factor (vWF) via their glycoprotein Ib-V-IX (GPIb-V-IX) complex. This initial adhesion is then stabilized by the direct interaction between collagen and platelet GPVI or GPIaIIa. Platelet adhesion can also be stabilized indirectly through the action of the vWF bound to collagen, which is then sensed by the GPIb-V-IX complex. Platelet adhesion then triggers multiple signaling pathways, resulting in platelet activation through the degranulation of numerous factors, including adenosine diphosphate (ADP), thromboxane A2 (TxA2) and serotonin (5-HT), through the P2Y12, TP and 5-HT2A receptors, respectively. Aside from degranulation, GPIIbIIIa undergoes a conformational change to its active form and binds to fibrinogen. The platelet secretion of hemostatic factors and their binding to fibrinogen fuel the autocrine activation loop of platelets, leading to aggregation and coagulation processes.

**Figure 3 ijms-22-07894-f003:**
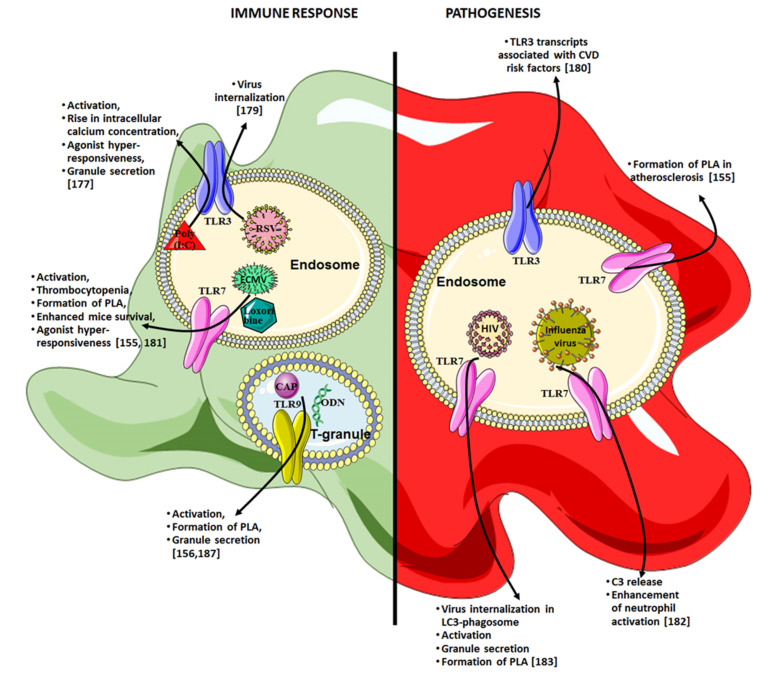
Platelet intracellular toll-like receptors involved in the immune response and pathogenesis. CAP: carboxyethylpyrrole protein adduct, HIV: human immunodeficiency virus, ODN: oligodeoxynucleotides, PLA: platelet-leukocyte aggregate, RSV: respiratory syncytial virus, TLR: toll-like receptor.

**Table 1 ijms-22-07894-t001:** Platelet response to stress.

Platelet Response to Stress	Major Biological Molecules Involved	Mechanisms/Consequences
Apoptosis	Death domains receptors Caspase activation cascades	Associated with platelet concentrate storage lesion [37] Cytoplasmic condensation [38] PS exposure [38] CD62P expression on the platelet plasma membrane [38] Altered mitochondrial signaling pathways, inner mitochondrial membrane disruption, and cytochrome c leaking [39]
	PKA	PKA activity correlated with platelet survival [40] Sequestration of prosurvival protein Bcl-xL in mitochondria upon PKA inhibition [41] Protection from apoptosis upon PKA activation [41]
	Siglec-7	Depolarization of the mitochondrial inner membrane [42] Overexpression of proapoptotic Bax and Bak proteins [42] Decreased expression of anti-apoptotic Bcl-2 proteins [42] Abnormal PS exposure to the platelet membrane [42] Formation of platelet microparticles [42]
	TRAF-2/5 FADD DEDAF	No activation of caspase 3/7 or PS exposure upon triggering of extrinsic apoptosis pathways [46]
	PAR-1 GPIIb/IIIa GPIbα	Platelet activation [48]
	TLR2	Caspase-3 activation [53] Bcl-xL degradation [54]
Necrosis	CypD	Formation of mitochondrial permeability transition pores in the inner mitochondria membrane [58,59] Contribution to thrombosis by displaying a procoagulant surface [60] Platelet association to neutrophils and accumulation in the brain [61] and in the lungs [57]
	DAMPs	Recruitment of proinflammatory phagocytes upon DAMP release by necrotic platelets [55]
	RIPK3	Thrombus formation and stability [67]
Autophagy	PI3k/Akt/mTOR pathway LC3-II ATG7 VPS34	Platelet adhesion and aggregation [72] Platelet activation related to autophagic flux [73] ATG7-dependent platelet aggregation, but ATG7-independent platelet count, fibrinogen binding and spreading [73] VPS34-dependent thrombus formation [74] Increased phenotypic platelet activation and decreased platelet aggregation rate [77] Negatively correlated with clinical platelet transfusion efficacy [77]

ATG: autophagy-related protein, Bak: Bcl-2 homologous antagonist killer, Bax: Bcl-2-associated X, Bcl-2: B-cell lymphoma 2, Bcl-xL: B-cell lymphoma extra-large, CypD: Cyclophilin D, DAMP: damage-associated molecular pattern, DEDAF: death effector domain-associated factor, FADD: Fas-associated protein with death domain, GP: glycoprotein, LC3: microtubule-associated proteins 1A/1B light chain 3, PAR-1: protease-activated receptor-1, PKA: protein kinase A, PS: phosphatidylserines, RIPK3: receptor-interacting protein kinase 3, Siglec: sialic acid-binding immunoglobulin-type lectin, TLR: toll-like receptor, TRAF: tumor necrosis factor (TNF)-associated factor, VPS: vacuolar protein sorting.

**Table 2 ijms-22-07894-t002:** Platelets as modulators of immune cell responses.

Immune Cells Interacting with Platelets.	Immunomodulatory Molecules Released by Platelet	Modulation of Immune Response
Neutrophils	Soluble CD62P and CD40L [86,87] Human β-defensing 1 [79]	Recruitment of neutrophils to inflammatory sites through interactions between neutrophil CD11b/CD18 and platelet JAM-3 or GPIbα [102,103,106] IL-8 release by neutrophils upon PSGL-1 activation [101] Formation of neutrophil extracellular traps [79,107] Reciprocal activation between neutrophils and platelets [108]
Monocytes	HMGB1 [65] Soluble CD40L [111]	Heterotypic platelet–monocyte aggregate formation through CD40/CD40L and PSGL-1/CD62P interactions and recruitment of monocytes to inflammatory sites [98,111] Monocyte activation and prevention of their apoptosis [65] Orientation to the pro-inflammatory monocyte response [109] Promotion of anti-inflammatory response of monocytes exposed to infectious stimuli [110]
Dendritic cells	Soluble CD40L [115]	Improvement of bacteria uptake and killing [115] Modulation of dendritic cell maturation [6,114] Promotion of antigen presentation to T cells [113]
B and T cells	CXCL4 [118] Soluble CD40 and CD40L [121]	Promotion of T cell trafficking [118] Modulation of B cell proliferation, differentiation, isotype switching and production [121,122]

CD40L: CD40 ligand, CXCL4: chemokine (C-X-C motif) ligand, HMGB1: high-mobility group box 1, IL: interleukin, JAM-3: junctional adhesion molecule 3, PSGL-1: P-selectin glycoprotein ligand-1.

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
