# Peer review of "Platelet Innate Immune Receptors and TLRs: A Double-Edged Sword"

_ijms, 2021, doi:10.3390/ijms22157894_

Round 1

Reviewer 1 Report

In my opinion, the most important pre-requisite for writing a good review article and getting it published in a high-quality journal are the authors themselves. They must have carried out comprehensive and exhaustive research on the topic covered by the subject of the manuscript. We are glad to see that this is exactly the case with this manuscript and its co-authors. A manuscript represents a critical, constructive analysis of platelet innate immune receptors and TLRs: a double-edged sword.

Suggestions for edition as well as some comments are the following:

Please rewrite the abstract to show the most relevant information about the platelet functions, their ability to respond to stress signals, to interact with immune cells and to modulate immune responses

Please change this keyword “Platelet” because it is presented in the title

However, systematic reviews (like this one) usually have a methods section. This section enables motivated researches to repeat the review. If for any reason, the authors would like to avoid a separate materials and methods section, then they should include some information about applied methods at the end of the introduction. The information should contain data sources (e.g. bibliographic databases), search terms and search strategies, selection criteria (inclusion/exclusion of studies), the number of studies screened and the number of studies included etc. Please provide this info to the readers.

Please include a figure about the central role of platelets in hemostasis

Please include a Table about platelet response to stress showing the most relevant information

Please include a Table about platelets as immune cells showing the most relevant information

Please include a Table about platelet contribution to pathogenesis through their innate immunity receptors showing the most relevant information

There are a lot of old references. Please update or remove references before 2013

I hope that my comments can improve the manuscript.

Author Response

Dear Icey, Dr Manoury, Editorial board members and Reviewers,

Thank you for considering our work, by T. Ebermeyer et al. (Manuscript # ijms-1297311), for publication in the International Journal of Molecular Sciences. We are grateful for the reviewers for their extensive reviewing and the time they spent on it. We are now pleased to submit a revised version of our manuscript -  "Please see the attachment."

Reviewer 2 Report

Review of “Platelet innate immune receptors and TLRs: a double-edged sword” (ijms-1297311)

This review articles summarized the new aspect of platelet. This review is interesting and meaningful. This reviewer has some concerns.

  1. How about the association between “Platelet response to stress” and “Platelets as immune cells”. This reviewer thinks that there is a possibility that platelets as immune cells may be responding to stress.
  2. Figure 1. The pathogenesis side. It's a little hard to see the black text on the red background.

Author Response

(The authors gave the same response as above.)

Round 2

Reviewer 1 Report

The manuscript was greatly improved and now is suitable for publication.